# An integrated bioinformatic investigation of kallikrein gene family members in kidney renel cell carcinoma

**Baoquan Wang**[1], **Lun Yang**[2], **Haiyun Qin**[2], **Fengzhen Li**[2]*, **Peitong Zhang**[1]*

1 Department of Oncology, Guang'anmen Hospital, China Academy of Chinese Medical Sciences, Beijing, China, 2 The Second Affiliated Hospital of Liaoning University of Traditional Chinese Medicine, Shenyang, China

* Lifz6072@163.com (FL); gamzpt@163.com (PZ)

**Data Availability Statement:** All relevant data are available from the Open Science Framework (https://osf.io/bnq4m/).

**Funding:** This work was supported by National Natural Science Foundation of China (Grant

## Abstract

### Backgrounds

*KLKs* have been proved to be key regulators of the tumor microenvironment. In this study, we explored the potential of *Kallikrein-related peptidases* (*KLKs*) as clinical diagnostic and prognostic markers in patients with kidney renal clear cell carcinoma (KIRC) as well as their relationship with common immuno-inhibitor and immune cell infiltration in the tumor micro-environment to provide new targets and novel ideas for KIRC therapy.

### Methods

Oncomine, Gene Expression Profiling Interactive Analysis (GEPIA), UCSC Xena, Genotype-Tissue Expression (GTEx), Kaplan-Meier plotter, cBioPortal, STRING, GeneMANIA, and TISIDB were used to analyze the differential expression, prognostic value, gene changes, molecular interaction, and immune infiltration of *KLKs* in patients with KIRC.

### Results

From the gene expression level, it can be determined that *KLK1*, *KLK6*, and *KLK7* are differentially expressed in KIRC and normal tissues. From the perspective of clinical prognosis, *KLK1*, *KLK13*, and *KLK14* are highly correlated with the clinical prognosis of KIRC. The expression of *KLKs* is regulated by various immunosuppressive agents, with KDR, PVRL2, and VTCN1 being the most significant. The expression of *KLKs* is significantly correlated with the infiltration of various immune cells, of which Eosinophils and Neutrophils are the most significant.

### Conclusions

*KLK1*, *KLK6*, *KLK7*, *KLK13*, and *KLK14* have potential as diagnostic and prognostic bio-markers, among which *KLK1* is the most significant. This study may provide detailed immune information and promising targets for KIRC immunotherapy to assist in designing new immunotherapies.

No.81673797). And the funders had no role in study design, data collection and analysis, decision topublish, or preparation of the manuscript.

**Competing interests:** The authors have declared that no competing interests exist.

## 1. Introduction

Kallikrein-related peptidases (*KLKs*), expressed in almost all human tissues, are a single family of 15 highly conserved trypsin or chymotrypsin-like serine proteases encoded by the largest uninterrupted protease gene (*KLK1-15*) in the human genome [1]. *KLKs* are secreted in the form of inactive zymogens, which are activated outside the cell by removing their propeptides. Then, *KLKs* participate in a series of proteolysis reactions and regulate important normal and pathobiological processes, such as the production or inactivation of peptide agonists from pre-cursor proteins, the release of membrane growth factor receptor agonists, and the activation or inactivation of growth factor receptors [2].

In terms of the tumor, *KLKs* have been proved to be key regulators of the tumor microenvironment. The interference and downstream signaling of the proteolysis cascade produced by these peptidases underlie tumorigenesis or inhibition of tumor growth [3]. For example, *KLKs* have proteolytic activity on extracellular matrix (ECM) proteins, cell membrane binding receptors, cell adhesion proteins, growth factors, and signal molecules, thus promoting the spread of cancer cells through their effects on cell migration and tissue invasion [4]. *KLKs* have significant potential as mediators of cancer progression, biomarkers of disease, and candidate targets for treatment [5]. Numerous studies have been conducted in related areas, such as ovarian cancer, breast cancer, prostate cancer, lung cancer, and skin cancer. For example, it is known that *KLK3*/PSA has been widely used in clinical practice as a biomarker of prostate cancer. Cancer vaccines and immunotherapies targeting *KLKs* have also achieved good results in clinical practice [6]. In terms of renal cell carcinoma (RCC), clinical experimental studies have proven that some *KLKs*, such as *KLK1*, *KLK3*, *KLK6*, *KLK7*, and *KLK15*, are differentially expressed in different subtypes of RCC, and *KLK6* has predictive value in RCC [7, 8].

RCC is a malignant tumor originating from the urinary tubular epithelial system of the renal parenchyma, accounting for about 3% of all cancers and 80%-90% of malignant renal tumors worldwide [9]. According to the statistics of the WHO, in 2020, the number of new cases of RCC was about 430,000, and the number of deaths was about 170,000 globally, ranking second in the incidence of urinary tract tumors. Kidney renal cell carcinoma (KIRC) is the most common RCC, with approximately 75% of RCC being KIRC and the highest fatality rate of all subtypes. It is of great clinical significance to explore the biomarkers and potential therapeutic targets of KIRC. Some studies have preliminarily shown that *KLKs* have potential as a tool for the diagnosis and prognosis of KIRC [10, 11]. Therefore, based on several large databases, we comprehensively analyzed the differential expression, potential function, the prognostic value of the *KLKs* gene family in KIRC, and its relationship with immune cell infiltration and immuno-inhibitor, and verified the previous conclusions in detail.

## 2. Materials and methods

### 2.1 Data collection

In addition to using various online databases, we also collected RNA-seq data and clinical information from TCGA (https://portal.gdc.cancer.gov/) and UCSC Xena [12] (https://xenabrowser.net). The UCSC Xena processes the data from TCGA through the Toil process, and includes normal human kidney RNA-seq data from the GTEx [13] (https://gtexportal.org/home), so we used it in gene differential expression section. In the analysis of gene differential expression, we collected RNA-seq data of kidney tissues from KIRC patients and normal human from UCSC, and the data from the same patient was excluded. This part included 531 KIRC samples, 72 paracancerous samples and 28 normal human kidney samples. In the part of prognostic analysis, our data collection criteria was the KIRC samples from TCGA should

have complete clinical data (including tumor stage, sex, age, total survival time). This part included 537 samples. In the part of immune infiltration analysis, our data collection criteria was the KIRC samples should have complete RNA-seq of 24 immune cells markers, and the duplicated RNA-seq data was excluded. This part included 530 samples. In the part of enrichment analysis, in order to explore the mechanism of key genes in KIRC patients, our data collection criteria was the KIRC samples should have complete RNA-seq, and the duplicated gene name data would be excluded in single-gene GSEA analysis. This part included 541 samples.

## 2.2 Oncomine

The transcription levels of *KLKs* in diverse cancer types were determined through analysis in Oncomine [14] (https://www.oncomine.org/resource/login.html), a publicly accessible online cancer microarray database. In the study, the expressions of *KLKs* were compared with normal controls from pan-cancer to KIRC in different clinical cancer specimens. A student test was used to determine whether the results were statistically significant. It was considered that "p-value < 0.05 and fold change > 2" was of existential significance.

## 2.3 GEPIA

GEPIA [15] (http://gepia.cancer-pku.cn) is an analytical web server that can dynamically analyze and visualize TCGA gene expression profile data, including thousands of normal and tumor tissue samples data. In this study, we used it to analyze the differential expression of *KLKs* in KIRC and normal tissues and the relationship between different expression levels of *KLKs* and clinical stages of KIRC.

## 2.4 Kaplan-Meier Plotter

Kaplan-Meier Plotter [16] (https://kmplot.com/analysis) includes data on about 54,000 genes and 21 cancer types with significant advantages in tumor survival analysis. We used the Kaplan-Meier Plotter to detect the prognostic value of different *KLKs* in KIRC patients, providing information about the relationship between gene expression and survival for patients with diverse cancers. In order to analyze the overall survival (OS) of KIRC patients, patient samples were divided into two groups by auto select best cut-off in Kaplan-Meier Plotter (high expression and low expression). The evaluated Kaplan Meier survival chart included the risk ratio (HR), 95% confidence interval (CI), and log-rank p-value.

## 2.5 cBioPortal

Based on the TCGA database, cBioPortal [17] (https://www.cbioportal.org) can visually analyze various cancers across genes, samples, and data types online and explore a wide range of multi-dimensional cancer genomes data. In this study, cBioPortal was used to analyze the gene mutations and related types of *KLKs* and to determine the degree of internal correlation.

## 2.6 GeneMANIA

GeneMANIA [18] (http://www.genemania.org) is a powerful website that uses highly accurate prediction algorithm to analysis gene lists and prioritize genes. We used GeneMANIA to represent the weight of *KLKs* physiological function prediction.

## 2.7 STRING

STRING [19] (https://version-11-5.string-db.org/) is an online database for searching known proteins and predicting protein-protein interactions. It synthesizes the data from various

databases to speculate the direct physical interaction between proteins and the indirect function correlation from the sources, such as experimental verification, gene proximity, co-expression, and chromosome proximity. Through the PPI network analysis in STRING, we predicted the interaction between *KLKs* and other molecules and performed a cluster analysis.

## 2.8 Enrichment analysis

Gene Ontology (GO) enrichment analysis (BP: biological process; CC: cellular component; MF: molecular function) and Kyoto Encyclopedia of Genes and Genomes (KEGG) pathway analysis was performed on the PPI network analysis results of STRING. We used DESeq2 and clusterProfiler R package to examine single-gene GSEA analysis. "FDR(qvalue)<0.25 and p. adjust<0.05" were used as the threshold to filter pathways.

## 2.9 TISIDB

TISIDB [20] (http://cis.hku.hk/TISIDB/index.php) is a newly developed database focusing on the interaction between tumors and immunity. It identifies genes related to tumor immune cell infiltration through high-throughput screening and genome analysis data. Additionally, it pre-calculates the association between genes and immune characteristics, such as lymphocytes, immunomodulators, and chemokines. In this study, TISIDB was used to predict the relationship between *KLKs* and immunomodulators in KIRC patients.

## 2.10 Immune infiltration assessment

GSVA package [21] was used to evaluate the immune infiltration of *KLKs* in KIRC, the details of immune cell markers was from previous literature [22, 23]. Spearman correlation was used to analyze the relationship between genes and immune cell infiltration. It is considered that the difference is statistically significant when $p < 0.05$.

## 2.11 Statistical analysis

In addition to the online analysis database, RStudio [24, 25] was used to analyze and visualize the downloaded data. Wilcoxon rank sum test was used to analyze the difference of gene expression. COX regression method was used to analyze the difference in prognosis when the data satisfied Proportional hazards hypothesis. The correlation of gene expression and immune cell infiltration was evaluated by Spearman method. $P<0.05$ was defined as statistically significant.

## 3. Results

### 3.1 Transcriptional levels of *KLKs* in patients with KIRC

Fifteen *KLK* factors are identified in mammals. Oncomine database was used to analyze the transcriptional level of *KLKs* in Kidney Cancer compared with normal tissues (Fig 1). The transcriptional changes of *KLKs* in different RCC subtypes were analyzed in detail, and the differential expression multiple, p-value, t-value, and data source were recorded (Table 1). According to the Oncomine database, in KIRC, the expression levels of *KLK1*, *KLK6*, *KLK7*, *KLK13*, and *KLK14* are down-regulated, while the expression of *KLK2* is up-regulated. *KLK5*, *KLK8*, *KLK9*, *KLK10*, and *KLK11* do not include RCC-related data, and *KLK3*, *KLK4*, *KLK12*, and *KLK15* do not include KIRC-related data.

In order to further analyze the differential expression of *KLKs* in KIRC, we first used the GEPIA database for analysis (Fig 2A). The results show that the expressions of *KLK1*, *KLK6*, and *KLK7* in KIRC are significantly lower than those in normal tissues ($p < 0.01$), but there is

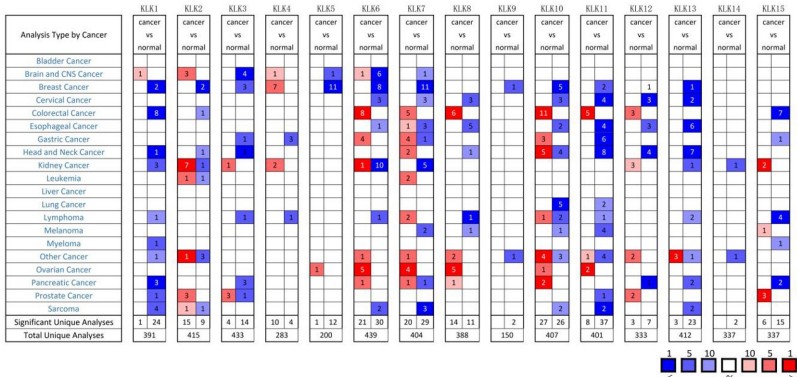

**Fig 1. The transcription levels of *KLKs* in different cancers.**

no significant difference in the expression of other genes. We downloaded data from Xena to analyze *KLKs* differential expression (Fig 2B) by Wilcoxon rank sum test on R software. The expressions of *KLK1*, *KLK3*, *KLK4*, *KLK5*, *KLK6*, *KLK7*, *KLK8*, *KLK10*, *KLK11*, and *KLK15* in KIRC are significantly lower than those in normal tissues (p < 0.001), while the expression of *KLK14* is up-regulated (p<0.05).

By combining the results of gene differential expression analysis in multiple databases, we took intersection of the results from different database, it can be confirmed that the expressions of *KLK1*, *KLK6*, and *KLK7* are significantly down-regulated in KIRC and normal tissues. The above factors have significant potential as biomarkers for the diagnosis of KIRC. The expressions of *KLK3*, *KLK5*, *KLK8*, *KLK10*, *KLK11*, and *KLK15* in Xena data are significantly down-regulated. However, due to the differences in statistical methods and the number of data cases, positive results are not obtained in the GEPIA database. The remaining results have conflicts in multiple databases, and further research is required to draw a definite conclusion.

## 3.2 Association of differential mRNA expression of *KLKs* with pathological parameters and prognosis of KIRC

Using the GEPIA dataset, we analyzed the expression of *KLKs* in the tumor stage of KIRC (Fig 3). The results indicate significant differences in the expressions of *KLK1*, *KLK13*, and *KLK14* in different clinical stages of KIRC. This result suggests that *KLK1*, *KLK13*, and *KLK14* might play an important role in the occurrence and development of KIRC. There is no significant difference among the other groups. More patient data need to be included to further verify the conclusion.

We further explored the influence efficiency of *KLKs* in the survival of patients with KIRC. Kaplan-Meier Plotter database was used to analyze OS (Fig 4A). The results show that low transcription of *KLK1*, *KLK2*, *KLK3*, *KLK8*, *KLK9*, *KLK10*, *KLK11*, *KLK12*, *KLK13*, and *KLK14* is significantly correlated with long OS (p<0.01); the high transcription of *KLK15* is significantly correlated with long OS (p<0.01). Then, the data from TCGA were adopted to analyze the relationship between differential expression of *KLKs* and clinical prognosis (Fig 4B), and the corresponding index was still OS. *KLK5*, *KLK8*, *KLK9*, *KLK11*, *KLK12*, and *KLK15* genes could not be grouped and analyzed due to their incomplete correlation data. The results show that the low expression of *KLK1*, *KLK2*, *KLK10*, *KLK13*, and *KLK14* genes is significantly correlated with long OS (p < 0.001).

**Table 1. The significant changes of *KLKs* expression in transcription level between different kidney cancers and normal kidney tissues.**

|  | Type of Kidney Cancer versus Normal Kidney Tissue | Fold Change | p Value | T Test | Source and/or Reference |
|---|---|---|---|---|---|
| KLK1 | Clear Cell Renal Cell Carcinoma | -6.233 | 2.34E-8 | -12.004 | Gumz Renal statistics [26] |
|  | Clear Cell Renal Cell Carcinoma | -2.766 | 5.29E-5 | -5.938 | Lenburg Renal statistics [27] |
|  | Renal Pelvis Urothelial Carcinoma | -2.433 | 1.15E-5 | -12.004 | Jones Renal statistics [28] |
| KLK2 | Clear Cell Renal Cell Carcinoma | 3.977 | 3.45E-33 | 35.261 | Jones Renal statistics [28] |
|  | Chromophobe Renal Cell Carcinoma | 3.728 | 2.88E-17 | 31.496 | Jones Renal statistics [28] |
|  | Papillary Renal Cell Carcinoma | 3.648 | 1.40E-16 | 24.282 | Jones Renal statistics [28] |
|  | Renal Oncocytoma | 3.434 | 2.53E-14 | 20.161 | Jones Renal statistics [28] |
|  | Chromophobe Renal Cell Carcinoma | 2.681 | 8.76E-4 | 4.946 | Yusenko Renal statistics [29] |
|  | Renal Wilms Tumor | 2.099 | 0.006 | 3.476 | Cutcliffe Renal statistics [30] |
|  | Clear Cell Sarcoma of the Kidney | 2.136 | 0.006 | 3.716 | Cutcliffe Renal statistics [30] |
|  | Renal Pelvis Urothelial Carcinoma | -2.198 | 1.78E-8 | -15.490 | Jones Renal statistics [28] |
| KLK3 | Chromophobe Renal Cell Carcinoma | 18.434 | 0.001 | 7.532 | Yusenko Renal statistics [29] |
| KLK4 | Renal Oncocytoma | 18.643 | 1.98E-4 | 6.792 | Yusenko Renal statistics [29] |
|  | Chromophobe Renal Cell Carcinoma | 15.100 | 0.002 | 5.198 | Yusenko Renal statistics [29] |
| KLK5 | NA | NA | NA | NA | NA |
| KLK6 | Renal Wilms Tumor | 3.158 | 1.33E-6 | 6.660 | Cutcliffe Renal statistics [30] |
|  | Papillary Renal Cell Carcinoma | -2.931 | 2.73E-21 | -24.592 | Jones Renal statistics [28] |
|  | Renal Oncocytoma | -3.190 | 1.36E-22 | -26.129 | Jones Renal statistics [28] |
|  | Clear Cell Renal Cell Carcinoma | -4.313 | 6.42E-25 | -24.492 | Jones Renal statistics [28] |
|  | Chromophobe Renal Cell Carcinoma | -5.517 | 1.68E-5 | -12.091 | Jones Renal statistics [28] |
|  | Clear Cell Renal Cell Carcinoma | -2.590 | 4.92E-7 | -7.277 | Gumz Renal statistics [26] |
|  | Clear Cell Sarcoma of the Kidney | -2.316 | 3.03E-5 | -8.603 | Cutcliffe Renal statistics [30] |
|  | Non-Hereditary Clear Cell Renal Cell Carcinoma | -4.926 | 4.10E-6 | -8.053 | Beroukhim Renal statistics [31] |
|  | Hereditary Clear Cell Renal Cell Carcinoma | -5.256 | 3.23E-6 | -8.445 | Beroukhim Renal statistics [31] |
|  | Clear Cell Renal Cell Carcinoma | -6.341 | 0.002 | -4.541 | Yusenko Renal statistics [29] |
|  | Papillary Renal Cell Carcinoma | -5.153 | 0.003 | -3.920 | Yusenko Renal statistics [29] |
| KLK7 | Clear Cell Renal Cell Carcinoma | -3.045 | 5.77E-2 | -31.723 | Jones Renal statistics [28] |
|  | Chromophobe Renal Cell Carcinoma | -2.681 | 7.04E-8 | -22.374 | Jones Renal statistics [28] |
|  | Clear Cell Sarcoma of the Kidney | -3.385 | 7.26E-6 | -6.525 | Cutcliffe Renal statistics [30] |
|  | Papillary Renal Cell Carcinoma | -2.900 | 0.004 | -3.486 | Yusenko Renal statistics [29] |
|  | Clear Cell Renal Cell Carcinoma | -2.487 | 0.007 | -3.575 | Yusenko Renal statistics [29] |
| KLK8 | NA | NA | NA | NA | NA |
| KLK9 | NA | NA | NA | NA | NA |
| KLK10 | NA | NA | NA | NA | NA |
| KLK11 | NA | NA | NA | NA | NA |
| KLK12 | Chromophobe Renal Cell Carcinoma | 3.760 | 0.003 | 3.840 | Yusenko Renal statistics [29] |
|  | Renal Wilms Tumor | 3.092 | 0.007 | 3.383 | Yusenko Renal statistics [29] |
|  | Renal Oncocytoma | 3.228 | 0.004 | 3.782 | Yusenko Renal statistics [29] |
| KLK13 | Clear Cell Renal Cell Carcinoma | -2.080 | 4.99E-6 | -6.083 | Gumz Renal statistics [26] |
| KLK14 | Clear Cell Renal Cell Carcinoma | -2.508 | 1.25E-5 | -5.704 | Gumz Renal statistics [26] |
| KLK15 | Chromophobe Renal Cell Carcinoma | 167.467 | 2.62E-6 | 12.414 | Yusenko Renal statistics [29] |
|  | Renal Oncocytoma | 25.045 | 7.60E-5 | 7.400 | Yusenko Renal statistics [29] |

NA, not available; TCGA, The Cancer Genome Atlas.

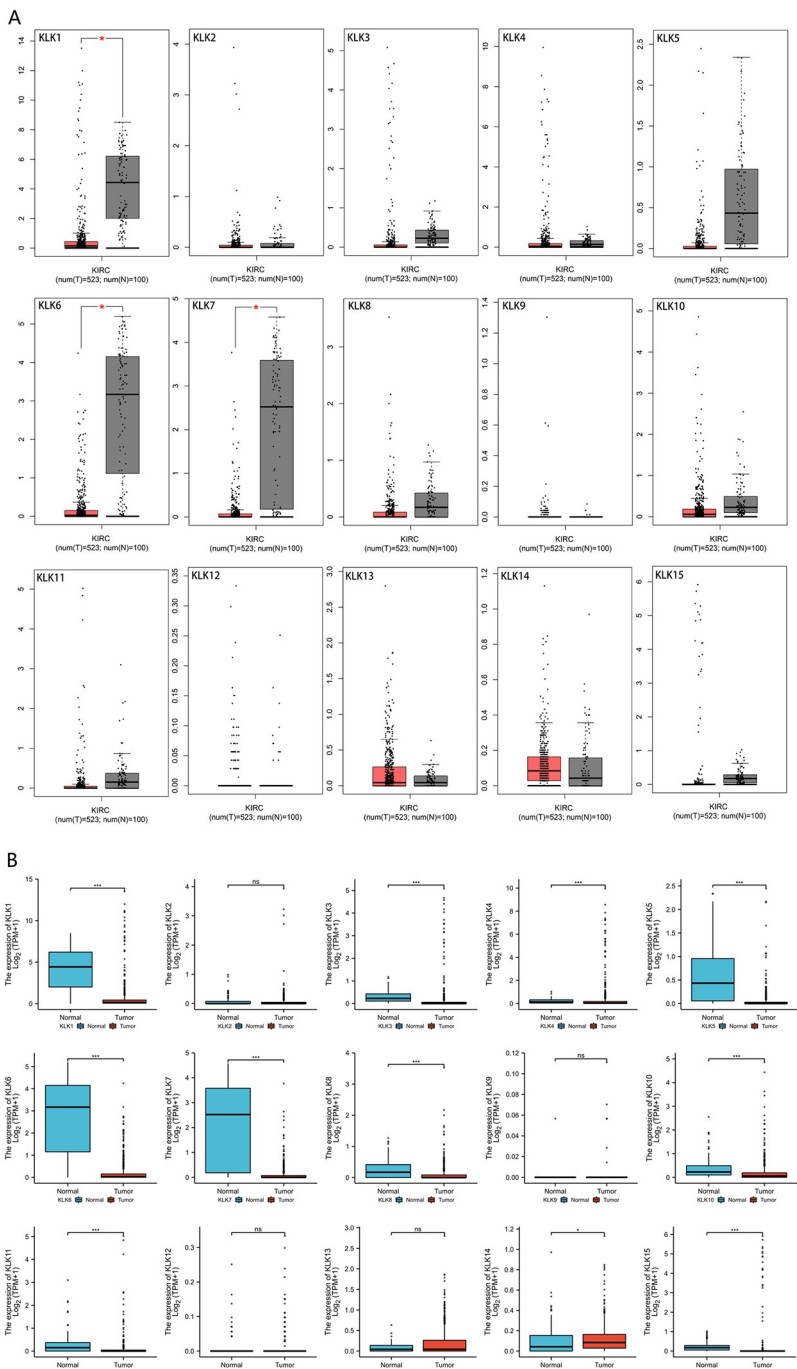

**Fig 2. The expression of *KLKs* in KIRC.**

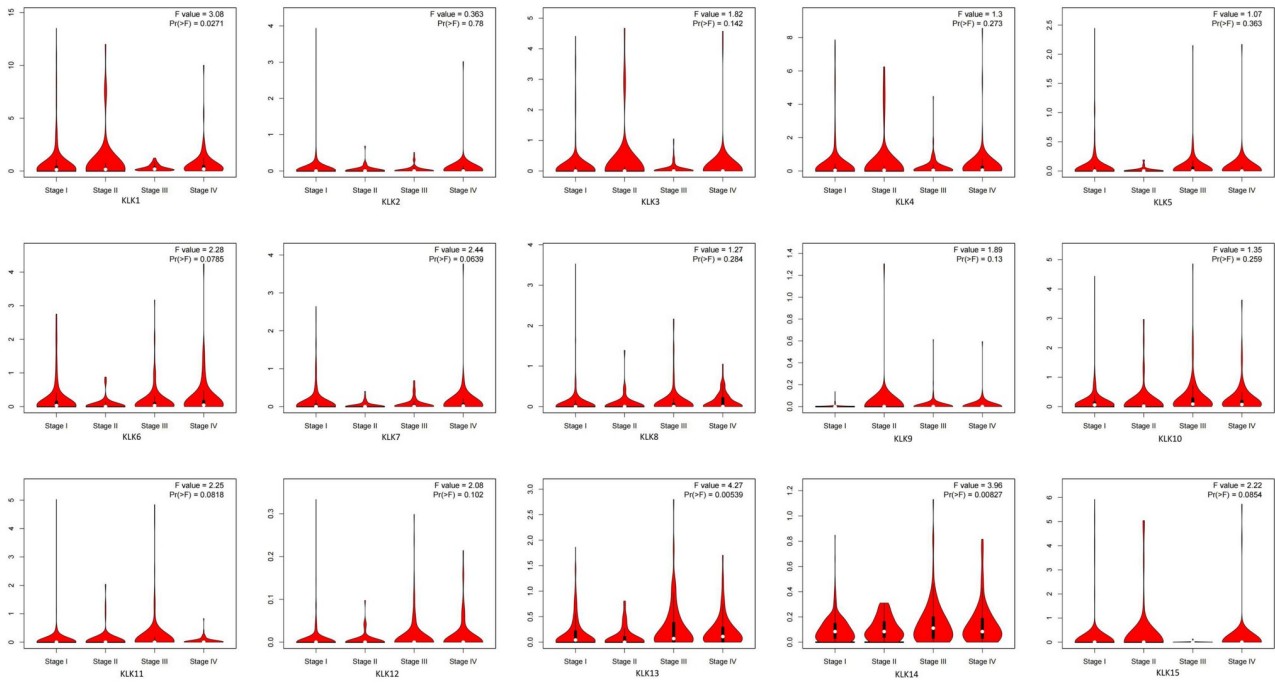

**Fig 3. Correlation between *KLKs* expression and tumor stages in KIRC patients.**

We took intersection of the clinical prognosis results from different database, it led to the conclusion that *KLK1*, *KLK13*, and *KLK14* are highly related to the clinical prognosis of KIRC.

### 3.3 Gene alteration, expression, and interaction analysis of *KLKs* in patients with KIRC

We used the cBioPortal tool to analyze the genetic changes of *KLKs* in patients with KIRC. Overall, 2 or more genetic changes were detected in KIRC patients with TCGA data sources, and mRNA high expression was more common in KIRC patients (Fig 5A). *KLKs* were altered in 118 samples of 538 KIRC patients, accounting for 22% (Fig 5B). The specific gene change frequency of *KLK1-15* in KIRC patients is shown in Fig 5B, with the mutation probability of *KLK1* being the highest (6%).

Moreover, a protein-protein interaction PPI network analysis of *KLKs* was conducted by STRING and GeneMANIA tools to explore their potential interactions. We inputted all the 15 genes of *KLKs* gene family into the two databases. There are 15 nodes and 6 edges in the primary PPI network of STRING. After online analysis, the first 20 genes that interact most closely with *KLKs* were selected in STRING. Cluster analysis was carried out in the STRING tool. As shown in the Fig 5C, the related genes are mainly divided into three categories in STRING. The main physiological functions of molecules were explored in the GeneMANIA tool, GeneMANIA results show that the functions of differentially expressed *KLKs* and their related molecules (such as PRSS3, PRSS37, PRSS56, and TMPRSS13) were mainly related to serine hydrolase activity, serine-type peptidase activity, antibacterial humoral response, and protein processing (Fig 5D). Using the cBioPortal online tool, we also analyzed the mRNA expression (RNA sequencing [RNA-seq] Version (V) 2RSEM) of *KLKs* in KIRC (TCGA) to calculate their correlation, including the correction of Pearson (Fig 5E). The results show that

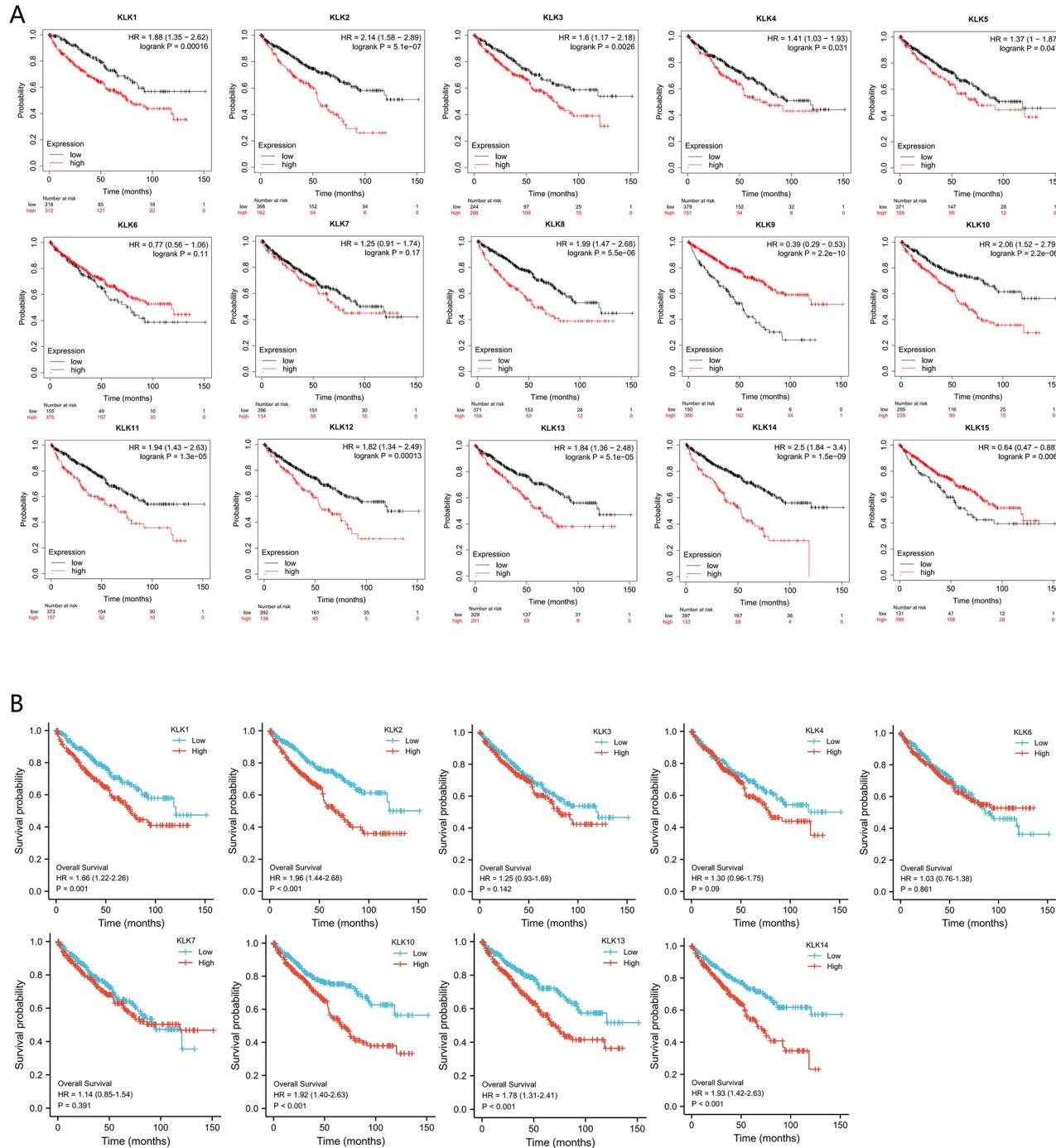

**Fig 4. The prognostic value of the mRNA level of *KLKs* in KIRC patients.**

there is a significant correlation existing among several groups of factors in *KLKs* (r > 0.25). And all factors except *KLK12* are correlated with other factors.

The 35 related genes obtained from the STRING online analysis tool in Fig 5C were enriched and analyzed by GO and KEGG in clusterProfiler package of R software. GO analysis predicted the function of target genes by three categories, the biological process (BP), cellular

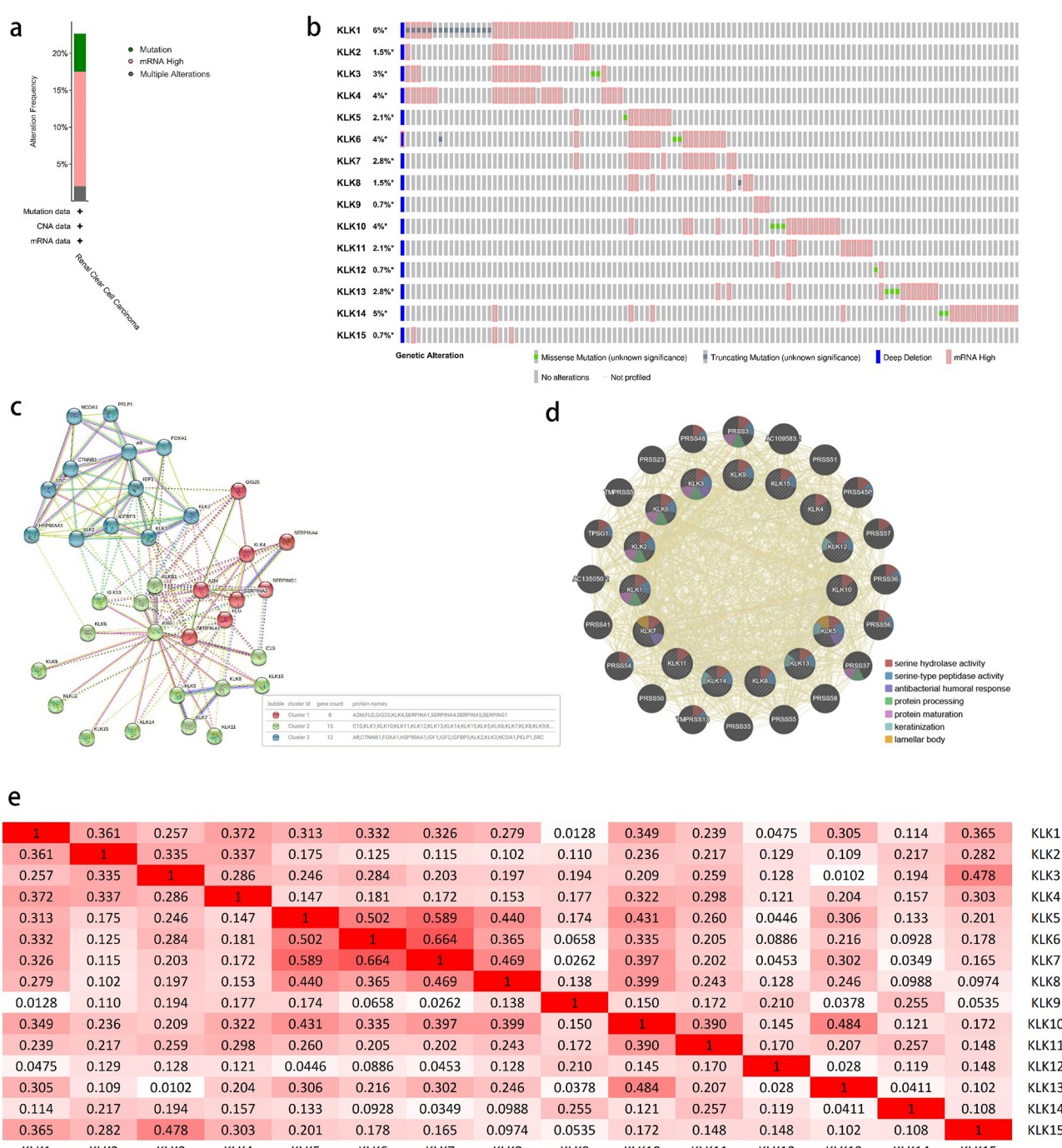

**Fig 5. Mutation, PPI network and function analysis of *KLKs* in KIRC.** (A, B) Summary of alterations in different expressed *KLKs*. (C-E) Protein-protein interaction network and correlation between different *KLKs*.

component (CC), and molecular function (MF). In this study, the top 5 genes were listed according to p-value values (S1 Table). The results are also shown in Fig 6. In BP and MF analysis, physiological processes such as proteolysis and blood coagulation are significantly regulated by *KLKs* changes, such as protein processing, protein maturation, platelet degranulation,

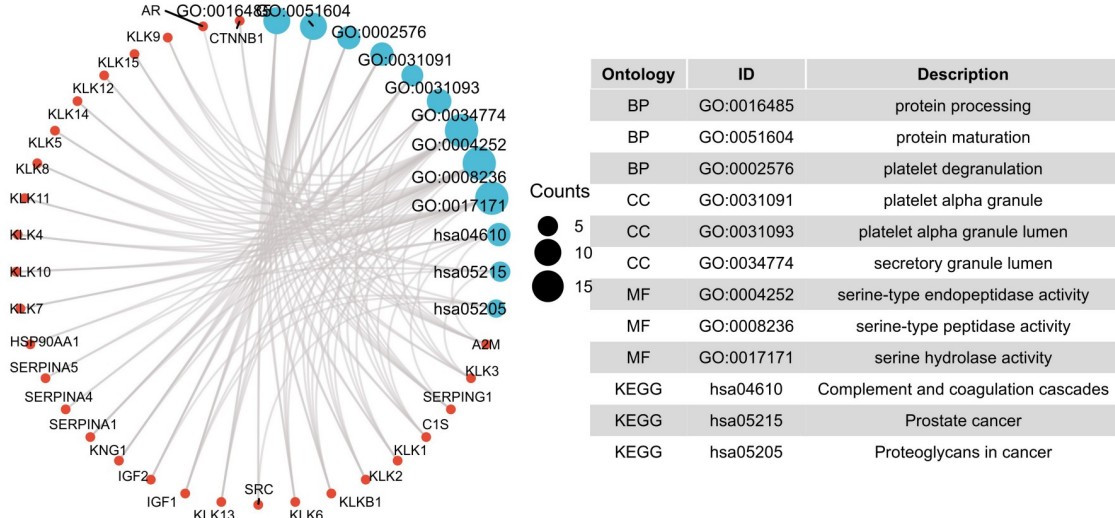

| Ontology | ID | Description |
|---|---|---|
| BP | GO:0016485 | protein processing |
| BP | GO:0051604 | protein maturation |
| BP | GO:0002576 | platelet degranulation |
| CC | GO:0031091 | platelet alpha granule |
| CC | GO:0031093 | platelet alpha granule lumen |
| CC | GO:0034774 | secretory granule lumen |
| MF | GO:0004252 | serine-type endopeptidase activity |
| MF | GO:0008236 | serine-type peptidase activity |
| MF | GO:0017171 | serine hydrolase activity |
| KEGG | hsa04610 | Complement and coagulation cascades |
| KEGG | hsa05215 | Prostate cancer |
| KEGG | hsa05205 | Proteoglycans in cancer |

**Fig 6. Results of GO and KEGG enrichment analysis.**

ECM disassembly, blood coagulation, intrinsic pathway, serine-type endopeptidase activity, serine-type peptidase activity, and serine hydrolase activity. CC analysis mainly involves platelet alpha granule, platelet alpha granule lumen, and vesicle lumen. KEGG analysis can identify pathways related to E2F alterations and adjacent genes that change frequently. By KEGG analysis, 18 pathways related to *KLKs* changes were found (S1 Table). Therefore, *KLKs* participate in the occurrence and development of KIRC via the above pathways, such as complement and coagulation, the rap1 signaling pathway, and the p53 signaling pathway. Among them, the rap1 signaling pathway and p53 signaling pathway are tumor suppressor gene-related pathways. Specifically, rap1 plays an important role in cell adhesion and integrin function of various cell types, thus participating in the invasion and metastasis of cancer [32]; p53 strictly regulates cell growth by promoting apoptosis and DNA repair. When p53 is mutated, it loses its function, leading to abnormal cell proliferation and tumor progression [33].

We also performed single-gene GSEA analysis to explore the possible pathway and mechanism of *KLK1*, *KLK6*, *KLK7*, *KLK13*, *KLK14* in KIRC (Fig 7). We found that "CD22 mediated BCR regulation" pathway was significantly enriched in *KLK1*,*KLK6* and *KLK13*, which was the most important pathway in the results. This suggested that "CD22 mediated BCR regulation" pathway may be the mechanism by which *KLKs* participates in the occurrence and development of KIRC. In addition, other important pathways in the results included: "Creation of C4 and C2 activators", "Transferrin endocytosis and recycling", "RNA Polymerase I Transcription".

## 3.4 The association between *KLKs* expression with immunoinhibitor and immune infiltration

In recent years, immunotherapy represented by immuno-inhibitor has achieved a breakthrough in the field of treatment [34]. Immuno-inhibitor can restore the anti-tumor immune response of hosts and induce tumor regression by blocking the negative immunomodulatory effect of immune checkpoints. Therefore, the relationship between *KLKs* family expression and immuno-inhibitor effect was studied using the TISIDB database. We collected positive results of immuno-inhibitor related to *KLKs* expression from the database. The result shows

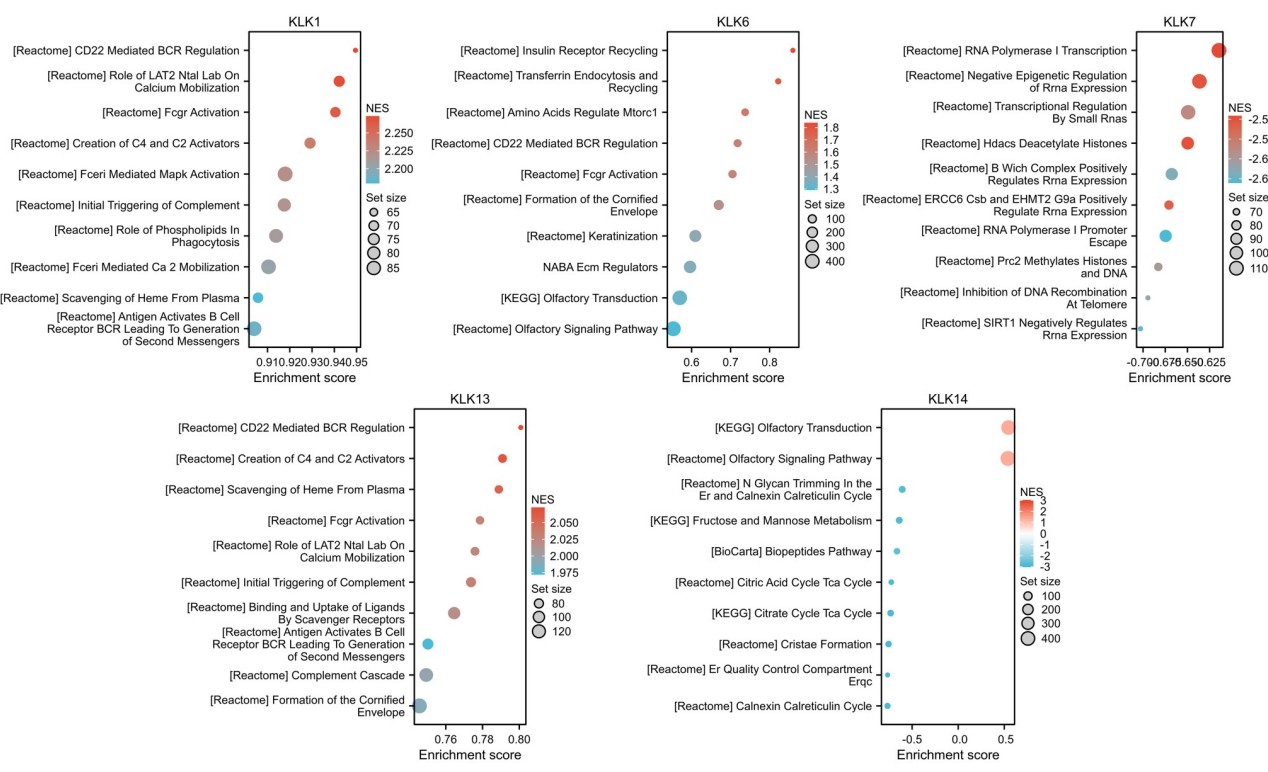

**Fig 7. Results of single-gene GSEA enrichment analysis.**

that *KLK1*, *KLK4*, *KLK5*, *KLK6*, *KLK7*, and *KLK10* are associated with multiple immuno-inhibitor (Table 2). KDR, PVRL2, VTCN1, CD274, IDO1, LGALS9, TGFBR1, CTLA4, LAG3, and HAVCR2 are most closely related to *KLKs*. KDR, CD274, IDO1, and HAVCR2 inhibit the expression of *KLKs*, but PVRL2, VTCN1, LGALS9, TGFBR1, CTLA4, and LAG3 promote the expression of *KLKs*.

The level of immune cells is related to the proliferation and development of cancer cells. The relationship between the infiltration of immune cells and the expression of *KLKs* in KIRC can be obtained from Fig 8A. It can be seen that 24 types of immune cell infiltration are related to *KLKs* in varying degrees, of which Eosinophils and Neutrophils are the most significant, and have a significant negative correlation with all *KLKs*, suggesting that these genes have an important role in the immune infiltration of KIRC. Then, we used Timer 2.0 (http://timer. cistrome.org/) to explore the relationship between the different expression of Eosinophils, Neutrophils and the prognosis of patients with KIRC, and further analyzed the same when we used *KLK1* as a subgroup (Fig 8B). The results show that high expression of Eosinophils and Neutrophil infiltration could improve the prognosis of patients, while different expression levels of *KLK1* could affect the effect of Eosinophils.

## 4. Discussion

The incidence of KIRC is on the rise worldwide [35]. It is a common malignant tumor of the urinary system. Approximately 70% of KIRC cases are diagnosed in the early stage, so they may be curable. The treatment is usually based on surgery, followed by regular follow-up. For patients with advanced KIRC, the primary clinical targeting drugs are tyrosine kinase inhibitors (TKIs), including inhibitors of the vascular endothelial growth factor (VEGF) pathway

**Table 2. The correlation between *KLKs* and immunoinhibitor.**

|  | KLK1 | KLK4 | KLK5 | KLK6 | KLK7 | KLK10 |
|---|---|---|---|---|---|---|
| KDR | -0.386 | -0.43 | -0.286 | -0.19 | -0.196 | -0.402 |
| PVRL2 | 0.211 | 0.233 | 0.23 | 0.24 | 0.203 | 0.269 |
| VTCN1 | 0.232 | 0.204 | 0.297 | 0.424 | 0.391 | 0.253 |
| CD274 | NA | -0.129 | -0.205 | -0.132 | -0.139 | -0.149 |
| IDO1 | -0.116 | -0.1 | -0.109 | -0.142 | -0.132 | NA |
| LGALS9 | 0.143 | NA | 0.177 | NA | 0.126 | 0.223 |
| TGFBR1 | 0.096 | NA | NA | 0.175 | 0.146 | NA |
| CTLA4 | 0.092 | NA | 0.115 | NA | NA | 0.174 |
| LAG3 | 0.135 | NA | 0.102 | NA | NA | 0.148 |
| HAVCR2 | NA | -0.129 | NA | -0.123 | NA | -0.185 |
| ADORA2A | -0.105 | -0.195 | NA | NA | NA | NA |
| TGFB1 | NA | NA | 0.102 | NA | NA | 0.149 |
| BTLA | 0.131 | NA | NA | NA | NA | 0.133 |
| CD96 | 0.087 | NA | NA | NA | NA | 0.122 |
| TIGIT | 0.096 | NA | NA | NA | NA | 0.117 |
| PDCD1 | 0.132 | NA | NA | NA | NA | 0.146 |
| PDCD1LG2 | NA | -0.176 | -0.103 | NA | NA | NA |
| IL10 | 0.091 | NA | NA | NA | NA | 0.107 |
| IL10RB | NA | NA | 0.085 | NA | NA | NA |
| CSF1R | NA | -0.115 | NA | NA | NA | NA |

NA: TISIDB shows no statistical significance or no related data.

and the mammalian target of rapamycin pathway (mTOR) [36, 37]. However, due to the high probability of tumor recurrence and metastasis, the prognosis of KIRC patients is usually poor. Therefore, there is an urgent need to develop new therapeutic strategies, including specific molecular targets, to reduce the mortality related to RCC.

In this study, multiple database analysis is used to verify the existing results, discarding the conflicting results from different database sources. In terms of the gene expression levels, there is a significant difference in the expression of *KLKs* in patients with KIRC. It can be determined that the expressions of *KLK1*, *KLK6*, and *KLK7* differ between KIRC and normal tissues, and the difference is statistically significant. From a perspective of clinical prognosis, the differential expression of the *KLKs* gene is also correlated with the prognosis of KIRC patients, and *KLK1*, *KLK13*, and *KLK14* are highly correlated with the clinical prognosis of KIRC. We took union set of the results of gene expression differences and prognosis, and the union result was used as potential biomarkers of early diagnosis and prognosis. And we found that *KLK1* exists in both gene lists. To sum up, in the *KLKs* gene family, *KLK1*, *KLK6*, *KLK7*, *KLK13*, and *KLK14* have the potential to serve as biomarkers for diagnosis and prognosis, with *KLK1* being the most significant.

*KLK1*, the first kallikrein-related peptidase discovered, is mainly present in urine, kidney, and pancreas. *KLK1* plays various beneficial roles in tissue injury protection with its anti-inflammatory, anti-apoptosis, anti-fibrosis, and antioxidant effects. In the existing studies, the use of *KLK1* as a target for treating cardiovascular, cerebrovascular, and renal diseases is a current hot spot [38]. In our study, the differential expression and clinical prognosis of *KLK1* in various databases are statistically significant, indicating the reliability of the results. Moreover, A previous study have confirmed that the expression of *KLK1* in KIRC is significantly lower

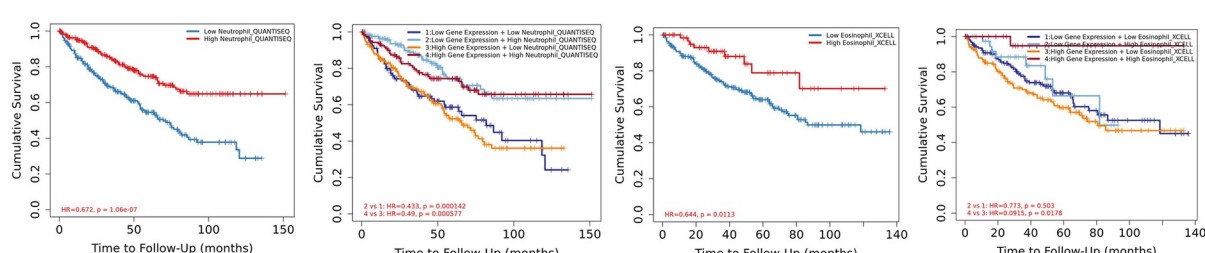

**Fig 8. Immune infiltration assessment.** (A) Correlation between differential expression of *KLKs* and immune cell infiltration. (B) Prognosis of patients with different expressions of Eosinophils, Neutrophils.

than that in normal tissues [10]. However, the study has indicated that *KLK1* is not statistically significant in the correlation analysis of clinical prognosis, which may be related to the small number of cases included.

To further evaluate the related functions of *KLKs*, we used STRING and GeneMANIA databases, and performed GO and KEGG enrichment analyses. In the molecular interaction analysis of the STRING database, cluster analysis shows that *KLK1*, *KLK6*, *KLK7*, *KLK13*, and *KLK14* are grouped into one group and closely interacted. According to the previous research results, the analysis of the GeneMANIA database, and the results of GO and KEGG enrichment analysis, *KLKs* are mainly involved in tumor growth, invasion, and metastasis by affecting proteolysis, degradation of ECM, treatment of growth factors and adhesion molecules, and regulation of apoptosis [39]. In addition, the results of single-gene GSEA analysis showed that the possible pathway of *KLK1* acting on KIRC is "CD22 mediated BCR regulation". Thus it may have an effect on the infiltration of immune cells.

Next, we investigated the relationship between *KLKs* expression and immuno-inhibitor. There is a significant negative correlation between KDR and most *KLKs*, while a significant positive correlation between PVRL2 and VTCN1. As a most critical factor in regulating angiogenesis, KDR is widely involved in tumor development and invasion [40, 41]. PVRL2 and VTCN1 have been studied in tumor-related immunotherapy by regulating the activity of immune cells. It has been confirmed that for many cancer patients, PVRL2 can change CD8 + T-cell cytokine production and cytotoxic activity [42]. The biological activity of VTCN1 is associated with inflammatory CD4+ T-cell responses and VTCN1- expressing tumor-associated macrophages and FoxP3+ regulatory T cells (T regs) within the tumor microenvironment [43]. Some studies have also shown that VTCN1 has lower expression levels in clear cell renal cell carcinoma [44]. In terms of immune infiltration, the expression of *KLKs* are significantly correlated with the infiltration of different immune cell types, of which Eosinophils and Neutrophils are the most significant. In the tumor microenvironment, immune cells have been proved to have the activity of promoting or inhibiting tumors. They are considered to be important determinants of clinical outcome and immunotherapy response. From the above immune-related information, it can be inferred that further studies on the relationship between *KLKs* and related immuno-inhibitor in KIRC can provide a promising target for KIRC immunotherapy and assist in the design of new immunotherapy.

In summary, *KLK1*, *KLK6*, *KLK7*, *KLK13*, and *KLK14* have the potential to be biomarkers for diagnosis and prognosis, with *KLK1* being the most significant. Moreover, this study may provide detailed immune information and promising targets for KIRC immunotherapy to assist in the design of new immunotherapies.

However, there are still some inevitable limitations in this study. The results of this study are mainly based on a number of large-scale online databases and have not been verified by experiments. These defects will be further remedied in our future research.

## Supporting information

**S1 Table. Results of GO KEGG enrichment analysis.**
(DOCX)

**S1 File.**
(ZIP)

**S1 Raw data.**
(ZIP)

**S1 Raw data.**
(ZIP)

## Acknowledgments

Thanks to Guang'anmen Hospital of China Academy of Chinese Medical Sciences and Liaoning Academy of Traditional Chinese Medicine for their contributions to this article.

## Author Contributions

**Data curation:** Baoquan Wang, Lun Yang.

**Formal analysis:** Baoquan Wang.

**Funding acquisition:** Peitong Zhang.

**Investigation:** Baoquan Wang.

**Methodology:** Baoquan Wang.

**Project administration:** Haiyun Qin.

**Software:** Baoquan Wang.

**Supervision:** Fengzhen Li.

**Validation:** Fengzhen Li, Peitong Zhang.

**Visualization:** Fengzhen Li, Peitong Zhang.

**Writing – original draft:** Baoquan Wang.

**Writing – review & editing:** Peitong Zhang.

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
