## [Decision Letter · Decision Letter 0]

14 Feb 2024

PONE-D-23-25080An Integrated Bioinformatic Investigation of Kallikrein Gene Family Members in Kidney Renel Cell CarcinomaPLOS ONE

Dear Dr. Zhang,

Thank you for submitting your manuscript to PLOS ONE. After careful consideration, we feel that it has merit but does not fully meet PLOS ONE’s publication criteria as it currently stands. Therefore, we invite you to submit a revised version of the manuscript that addresses the points raised during the review process.

We look forward to receiving your revised manuscript.

Kind regards,

Gurudeeban Selvaraj

Academic Editor

PLOS ONE

Journal Requirements:

Reviewers' comments:

Reviewer's Responses to Questions

**Comments to the Author**

1. Is the manuscript technically sound, and do the data support the conclusions?

Reviewer #1: Yes

2. Has the statistical analysis been performed appropriately and rigorously? 

Reviewer #1: Yes

3. Have the authors made all data underlying the findings in their manuscript fully available?

Reviewer #1: Yes

4. Is the manuscript presented in an intelligible fashion and written in standard English?

Reviewer #1: Yes

5. Review Comments to the Author

Reviewer #1: Authors have performed “An Integrated Bioinformatic Investigation of Kallikrein Gene Family Members in Kidney Renel Cell Carcinoma”, and conclude this study may provide detailed immune information and promising targets for KIRC immunotherapy to assist in designing new immune therapies, almost the maximum number of computational tools were utilized for this work and i have a few queries to ask,

1. Various series of analyses were performed to justify the work already reported. Or else it is said that new genes identified or key genes for the development of Kidney Renel Cell Carcinoma

2. Authors have to specify the inclusion criteria of the data collection

3. As mentioned in section 2.4, median expression (high expression and low expression) as well as log-rank P-value were mentioned. What were the actual cut-off values?

4. A separate method section for statistical analysis should be included in the manuscript

5. How many genes are given as input for STRING and GeneMANIA analysis, authors should specify the total number of edges and nodes in the primary PPI network

6. What were the criteria for hug gene identification?

7. I strongly recommend incorporating a discussion of significant pathways associated with the key genes in the Discussion sections. By doing so, you will provide a more comprehensive understanding of the functional implications of these genes

6. PLOS authors have the option to publish the peer review history of their article (what does this mean?). If published, this will include your full peer review and any attached files.

Reviewer #1: No

---

## [Author Response · Author response to Decision Letter 0]

6 Mar 2024

Dear Editors and Reviewers:

 Thank you for your letter and for the reviewer’s comments concerning our manuscript entitled ‘An Integrated Bioinformatic Investigation of Kallikrein Gene Family Members in Kidney Renel Cell Carcinoma’(ID:PONE-D-23-25080).Those comments are all valuable and very helpful for revising and improving our paper, as well as the important guiding significance to our researches. We have studied comments carefully and have made correction which we hope meet wtih approval. The revised portion about the manuscript are marked in blue in the cover letter. The main corrections in the paper and the point-by-point response to the reviewer’s comments are as follows:

Academic Edtior:

Response: Thank you for your reply. We have revised the manuscript.

2.Note from Emily Chenette, Editor in Chief of PLOS ONE, and Iain Hrynaszkiewicz, Director of Open Research Solutions at PLOS: Did you know that depositing data in a repository is associated with up to a 25% citation advantage (https://doi.org/10.1371/journal.pone.0230416)? If you’ve not already done so, consider depositing your raw data in a repository to ensure your work is read, appreciated and cited by the largest possible audience. You’ll also earn an Accessible Data icon on your published paper if you deposit your data in any participating repository (https://plos.org/open-science/open-data/#accessible-data).

Response: Thank you for your suggestion. We are willing to deposit raw data in a participating repository. And we have added the raw data of the new section named single-gene GSEA enrichment analysis in the "raw data" zip file and uploaded it.

3.Please note that funding information should not appear in any section or other areas of your manuscript. We will only publish funding information present in the Funding Statement section of the online submission form. Please remove any funding-related text from the manuscript.

Response: Thank you for your reply. We have revised the manuscript.

4.We note that the grant information you provided in the ‘Funding Information’ and ‘Financial Disclosure’ sections do not match. When you resubmit, please ensure that you provide the correct grant numbers for the awards you received for your study in the ‘Funding Information’ section.

Response: We are very sorry for our negligence about that. We ensure the grant numbers for the fund is correct, and update the statement of financial disclosure as following : 

"This work was supported by National Natural Science Foundation of China (Gant No.81673797). And the funders had no role in study design, data collection and analysis, decision topublish, or preparation of the manuscript."

Reviewer:

1. Various series of analyses were performed to justify the work already reported. Or else it is said that new genes identified or key genes for the development of Kidney Renel Cell Carcinoma

Response: Thanks for your kind comments. We tried our best to use as many analytical tools as we could to prove this finding and report the process. In the future, we will apply for other funding in order to conduct in vivo and in vitro experiments to explore the specific mechanism of what the KLKs participates in the occurrence and development of KIRC.

2. Authors have to specify the inclusion criteria of the data collection

Response: We think this is an excellent suggestion. We have supplemented the criteria of the data collection in the corresponding part, including the exact location where the change can be found in the revised manuscript. 

"2.1 Data collection

In addition to using various online databases, we also collected RNA-seq data and clinical information from TCGA (https://portal.gdc.cancer.gov/) and UCSC Xena [12] (https://xenabrowser.net). The UCSC Xena processes the data from TCGA through the Toil process, and includes normal human kidney RNA-seq data from the GTEx [13] (https://gtexportal.org/home), so we used it in gene differential expression section. In the analysis of gene differential expression, we collected RNA-seq data of kidney tissues from KIRC patients and normal human from UCSC, and the data from the same patient was excluded. This part included 531 KIRC samples, 72 paracancerous samples and 28 normal human kidney samples. In the part of prognostic analysis, our data collection criteria was the KIRC samples from TCGA should have complete clinical data (including tumor stage, sex, age, total survival time). This part included 537 samples. In the part of immune infiltration analysis, our data collection criteria was the KIRC samples should have complete RNA-seq of 24 immune cells markers, and the duplicated RNA-seq data was excluded. This part included 530 samples. In the part of enrichment analysis, in order to explore the mechanism of key genes in KIRC patients, our data collection criteria was the KIRC samples should have complete RNA-seq, and the duplicated gene name data would be excluded in single-gene GSEA analysis. This part included 541 samples."

3. As mentioned in section 2.4, median expression (high expression and low expression) as well as log-rank P-value were mentioned. What were the actual cut-off values?

Respones: We are very sorry for our negligence of the writing in this part. In Figure 4B, We used the COX regression (which all data satisfies the PH hypothesis) to draw the K-M curve, and the grouping method in this part is taking the median expression as the cut-off value. In Figure 4A which described in section 2.4, "divided into two groups as median expression" is misused. This is a mistake and has been revised in the corresponding part of the manuscript. The relevant parts have been revised. Thank you very much for pointing it out.

"patient samples were divided into two groups by auto select best cut-off in Kaplan-Meier Plotter (high expression and low expression)"

4. A separate method section for statistical analysis should be included in the manuscript

Response: Thank you for your suggestion. We have written 2.11 section to describe statistical analysis which we used in the manuscript.

"2.11 Statistical analysis

In addition to the online analysis database, RStudio [24,25] was used to analyze and visualize the downloaded data. Wilcoxon rank sum test was used to analyze the difference of gene expression. COX regression method was used to analyze the difference in prognosis when the data satisfied Proportional hazards hypothesis. The correlation of gene expression and immune cell infiltration was evaluated by Spearman method. P<0.05 was defined as statistically significant."

5. How many genes are given as input for STRING and GeneMANIA analysis, authors should specify the total number of edges and nodes in the primary PPI network

Response: Thank you for your suggestion. We inputted 15 genes of KLKs gene family into STRING (version11.5) and GeneMANIA database, and obtained the first 20 genes that interact most closely with KLKs. The primary PPI network has 15 nodes and 6 edges as shown in Figure 1A. The relevant parts of the manuscript have been supplemented. In STRING, the PPI network analysis process is as Figure 1 shown.

"We inputted all the 15 genes of KLKs gene family into the two databases. There are 15 nodes and 6 edges in the primary PPI network of STRING. After online analysis, the first 20 genes that interact most closely with KLKs were selected in STRING."

6. What were the criteria for hug gene identification?

Response: Thank you for your suggestion. In fact, we have adopted the following criteria for key gene identification in the KLKs gene family, and the relevant part has been added to the 3.1,3.2 section and Discussion.

1.In the analysis of different databases of gene expression differences, we took the intersection of KLKs gene family’s differential expression analysis results from different databases. This strategy was also applied to the analysis of prognosis (including different stages of the tumor). Because tumor stages are highly related to prognosis, this part was included.

2.Because the KLKs gene family has a high correlation in gene co-expression except KLK12, and in clinical practice, the biomarkers of early diagnosis and prognosis often overlap. Due to this reason, We took union set of the results of gene expression differences and prognosis, and the union result was used as potential biomarkers of early diagnosis and prognosis. We found that KLK1 exists in both gene lists, so it is the most significant.

"By combining the results of gene differential expression analysis in multiple databases, we took intersection of the results from different database, it can be confirmed that..." 

"We took intersection of the clinical prognosis results from different database, it led to the conclusion that..."

"We took union set of the results of gene expression differences and prognosis, and the union result was used as potential biomarkers of early diagnosis and prognosis. And we found that KLK1 exists in both gene lists. To sum up, in the..."

7.I strongly recommend incorporating a discussion of significant pathways associated with the key genes in the Discussion sections. By doing so, you will provide a more comprehensive understanding of the functional implications of these genes

Response: Thank you for your suggestion. In order to achieve this goal scientifically, we added the single-gene GSEA analysis of KLK1,KLK6,KLK7,KLK13,KLK14 to supplement the exploration of the pathways. The specific parts have been supplemented in the manuscript.

"Section 2.8

We used DESeq2, edgeR and clusterProfiler R package to examine single-gene GSEA analysis."FDR(qvalue)<0.25 and p.adjust<0.05" were used as the threshold to filter pathways."

"Section 3.3

We also performed single-gene GSEA analysis to explore the possible pathway and mechanism of KLK1, KLK6, KLK7, KLK13, KLK14 in KIRC (Figure 7). We found that ‘CD22 mediated BCR regulation’ pathway was significantly enriched in KLK1,KLK6 and KLK13, which was the most important pathway in the results. This suggested that ‘CD22 mediated BCR regulation’ pathway may be the mechanism by which KLKs participates in the occurrence and development of KIRC. In addition, other important pathways in the results included: ‘Creation of C4 and C2 activators’, ‘Transferrin endocytosis and recycling’, ‘RNA Polymerase I Transcription’."

"4.Discussion

In addition, the results of single-gene GSEA analysis showed that the possible pathway of KLK1 acting on KIRC is "CD22 mediated BCR regulation". Thus it may have an effect on the infiltration of immune cells."

Many thanks for your time and consideration.

---

## [Decision Letter · Decision Letter 1]

26 Apr 2024

PONE-D-23-25080R1An Integrated Bioinformatic Investigation of Kallikrein Gene Family Members in Kidney Renel Cell CarcinomaPLOS ONE

Dear Dr. Zhang,

Thank you for submitting your manuscript to PLOS ONE. After careful consideration, we feel that it has merit but does not fully meet PLOS ONE’s publication criteria as it currently stands. Therefore, we invite you to submit a revised version of the manuscript that addresses the points raised during the review process.

We look forward to receiving your revised manuscript.

Kind regards,

Gurudeeban Selvaraj

Academic Editor

PLOS ONE

Journal Requirements:

**Editor Comments**:

I strongly suggest that the authors need to deposit their codes for R packages DESeq2, edgeR and clusterProfiler etc in github or appendix.

Reviewers' comments:

Reviewer's Responses to Questions

**Comments to the Author**

1. If the authors have adequately addressed your comments raised in a previous round of review and you feel that this manuscript is now acceptable for publication, you may indicate that here to bypass the “Comments to the Author” section, enter your conflict of interest statement in the “Confidential to Editor” section, and submit your "Accept" recommendation.

Reviewer #1: All comments have been addressed

2. Is the manuscript technically sound, and do the data support the conclusions?

Reviewer #1: Yes

3. Has the statistical analysis been performed appropriately and rigorously? 

Reviewer #1: Yes

4. Have the authors made all data underlying the findings in their manuscript fully available?

Reviewer #1: Yes

5. Is the manuscript presented in an intelligible fashion and written in standard English?

Reviewer #1: Yes

6. Review Comments to the Author

Reviewer #1: (No Response)

7. PLOS authors have the option to publish the peer review history of their article (what does this mean?). If published, this will include your full peer review and any attached files.

Reviewer #1: No

---

## [Author Response · Author response to Decision Letter 1]

3 May 2024

Guang’anmen Hospital, China Academy of Chinese Medical Sciences 

Beijing, China, 5/03/2024

Dear Editors and Reviewers:

Thank you for your letter about our manuscript entitled ‘An Integrated Bioinformatic Investigation of Kallikrein Gene Family Members in Kidney Renel Cell Carcinoma’(ID:PONE-D-23-25080). Those comments are very helpful for improving our paper. We have improved the paper along the lines suggested by those comments which we hope meet wtih approval. The revised portion about the manuscript are marked in blue in this letter. The main corrections in the paper and the point-by-point response to comments are as follows:

Journal Requirements:

1.Please review your reference list to ensure that it is complete and correct. If you have cited papers that have been retracted, please include the rationale for doing so in the manuscript text, or remove these references and replace them with relevant current references. Any changes to the reference list should be mentioned in the rebuttal letter that accompanies your revised manuscript. If you need to cite a retracted article, indicate the article’s retracted status in the References list and also include a citation and full reference for the retraction notice.

Response: Thank you for your reply. We have revised the manuscript as follow.

"44.Jing X, Qin X, Liu H, Liu H, Wang H, Qin J, et al. DNA damage response alterations in clear cell renal cell carcinoma: clinical, molecular, and prognostic implications. Eur J Med Res. 2024 Feb 7;29(1):107. doi: 10.1186/s40001-024-01678-x."

Editor Comments:

1.I strongly suggest that the authors need to deposit their codes for R packages DESeq2, edgeR and clusterProfiler etc in github or appendix.

Response: Thank you for your suggestion. We have supplemented the relevant codes for R packages in zip file named "demo.R".

Many thanks for your time and consideration.

Yours Sincerely, 

Baoquan Wang

Guang’anmen Hospital, China Academy of Chinese Medical Sciences

Tel: +86 18810216061 Email: 20200931825@bucm.edu.cn

---

## [Editor Report · Decision Letter 2]

23 May 2024

An Integrated Bioinformatic Investigation of Kallikrein Gene Family Members in Kidney Renel Cell Carcinoma

PONE-D-23-25080R2

Dear Dr. Zhang,

We’re pleased to inform you that your manuscript has been judged scientifically suitable for publication and will be formally accepted for publication once it meets all outstanding technical requirements.

Kind regards,

Gurudeeban Selvaraj, Ph.D.

Academic Editor

PLOS ONE

---

## [Editor Report · Acceptance letter]

1 Jul 2024

PONE-D-23-25080R2 

PLOS ONE

Dear Dr. Zhang, 

I'm pleased to inform you that your manuscript has been deemed suitable for publication in PLOS ONE. Congratulations! Your manuscript is now being handed over to our production team.

Kind regards, 

on behalf of

Dr. Gurudeeban Selvaraj 

Academic Editor

PLOS ONE